# Assessment of *in vivo* bone microarchitecture changes in an anti-TNFα treated psoriatic arthritic patient

Enrico Soldati[1,2,3]*, Lucas Escoffier[4], Sophie Gabriel[5], Augustin C. Ogier[1,6],
Christophe Chagnaud[4], Jean P. Mattei[1,4], Serge Cammilleri[5], David Bendahan[1],
Sandrine Guis[1,4]

1 Aix-Marseille Université, CNRS, CRMBM-CEMEREM, Marseille, France, 2 Aix-Marseille Université,
CNRS, IUSTI, Marseille, France, 3 Aix-Marseille Université, CNRS, ISM, Marseille, France, 4 Aix-Marseille
Université, Service de Rhumatologie, AP-HM, Marseille, France, 5 Aix-Marseille Université, Service de
Médecine Nucléaire, AP-HM, Institut Fresnel, Marseille, France, 6 Aix-Marseille Université, Université de
Toulon, CNRS, LIS, Marseille, France

* enrico.soldati@univ-amu.fr

## Abstract

### Objective

Psoriatic arthritis (PsA) is an inflammatory rheumatic disease, mediated in part by TNFα
and associated with bone loss. Anti-TNFα treatment should inhibit this phenomenon and
reduce the systemic bone loss. Ultra-high field MRI (UHF MRI) may be used to quantify
bone microarchitecture (BM) *in-vivo*. In this study, we quantified BM using UHF MRI in a
PsA patient and followed up the changes related to anti-TNFα treatment.

### Subjects and methods

A non-treated PsA patient with knee arthritis and 7 gender-matched controls were scanned
using a gradient re-echo sequence at UHF MRI. After a year of Adalimumab treatment, the
patient underwent a second UHF MRI. A PET-FNa imaging was performed before and after
treatment to identify and localize the abnormal metabolic areas. BM was characterized
using typical morphological parameters quantified in 32 regions of interest (ROIs) located in
the patella, proximal tibia, and distal femur.

### Results

Before treatment, the BM parameters were statistically different from controls in 24/32 ROIs
with differences reaching up to 38%. After treatment, BM parameters were normalized for
15 out of 24 ROIs. The hypermetabolic areas disclosed by PET-FNa before the treatment
partly resumed after the treatment.

### Conclusion

Thanks to UHF MRI, we quantified *in vivo* BM anomalies in a PsA patient and we illustrated
a major reversion after one year of treatment. Moreover, BM results highlighted that the

journal.pone.0251788

Sciences in Lublin, POLAND

**Data Availability Statement:** According to the
restrictions imposed by Aix Marseille University
and by the local ethics committee regarding
patients data sharing, data could be made available

upon reasonable request addressed to Monique Bernard (monique.bernard@univ-amu.fr) pending the signature of a MTA approved by Aix Marseille University.

**Funding:** ES has received funding from the European Union's Horizon 2020 research and innovation program under the Marie Skodowska-Curie grant agreement No713750. Also, it has been carried out with the financial support of the Regional Council of Provence- Alpes-Côte d'Azur and with the financial support of the A∗MIDEX (n˚ ANR- 11-IDEX-0001-02), funded by the "Investissements d'Avenir" project funded by the French Government, managed by the French National Research Agency (ANR). The funders had no role in study design, data collection and analysis, decision to publish, or preparation of the manuscript.

**Competing interests:** The authors have declared that no competing interests exist.

abnormalities were not only localized in hypermetabolic regions identified by PET-FNa, suggesting that the bone loss was global and not related to inflammation.

## Introduction

Psoriatic arthritis (PsA) is an inflammatory rheumatic joint disease associated with psoriasis in which axial and peripheral joints can display an elevated inflammatory status [1]. PsA has been initially described by Moll and Wright as a seronegative inflammatory arthritis that occurs most of the time in the presence of psoriasis [2]. It was initially thought to be rare but recent studies indicated that it might occur in up to 30% of patients with psoriasis [3, 4]. The most commonly involved sites include Achilles tendon, quadriceps tendon, knee, wrist and ankle [5]. These sites are usually assessed using ultrasound imaging which could detect both clinically active and non-active sites. Most of the times sites are clinically active. The main clinical presentations are swollen, tender joints, stiffness and pain, scaly skin patches, nail pitting, eye redness [6] but also asymmetric oligo-arthritis, polyarthritis, dactylitis and enthesis [1, 7]. The PsA clinical presentation is frequently associated with structural changes such as bone erosion and formation i.e. ankylosis or periostitis [5, 8]. Bone erosion could lead to fragility fractures which is a relevant clinical event and one of the major complication of many bone disorders such as osteoporosis. While the prevalence of osteoporosis in PsA is still a matter of debate [9, 10], previous studies have shown that fragility fractures should be considered when evaluating the global picture of PsA patients [10]. Psoriasis and psoriatic arthritis are characterized by tissue infiltration by activated T cells thereby resulting in an increased TNFα, IL 17 and IL 23 production [7, 11, 12]. Synovial tissue and entheses are more particularly affected [13]. This pro inflammatory status can be an effective trigger of osteoclasts differentiation and activation through the expression of the receptor activator of nuclear factor kappa B ligand (RANKL) [14].

The increased cell activity and the corresponding elevated inflammatory status due to PsA could be assessed using positron emission tomography (PET), which is able to assess the abnormal accumulation of radiotracer in specific areas [15, 16]. The systemic bone loss resulting in a reduced bone mineral density (BMD) and the role of TNFα antibodies in this process are a matter of debate in psoriatic arthritis [8, 17–21]. Using dual energy X-ray absorptiometry (DXA) [22], reduced BMD (g/cm$^2$) values have been reported in PsA patients as compared to controls and so regardless of sex, menopausal status, or age (lumbar spine 1.112 *vs*. 1.326; femoral neck 0.870 *vs*. 1.006; total body 1.125 *vs*. 1.203) [23]. However, bone micro architecture has never been documented as part of this bone alteration process. Interestingly, magnetic resonance imaging (MRI) and more particularly ultra-high field MRI (UHF MRI) has been reported as a promising tool for the assessment of bone microarchitecture given the high resolution of the corresponding images [24]. Over the last few years, this non-radiating imaging technique has shown promising results regarding spine, knee, and femur trabeculation in osteoporosis [25–27]. So far, the corresponding changes in psoriatic arthritis have never been assessed.

The purpose of the present study was to investigate bone trabeculation in a patient with psoriatic arthritis using UHF MRI and to assess changes related to a TNFα antibodies therapeutic strategy.

## Material and methods

### Subject recruitment

This study received institutional review board approval by the "Comite de protection des personnes sud Méditerranée I" (approval number 2016-A000427-44). Written informed consent

was obtained from all the recruited subjects. One PsA patient (male, 18 years old, body mass index (BMI) = 14.53 kg/m$^2$) affected by axial and peripheral psoriatic arthritis, was assessed before and after a one-year Adalimumab treatment. The patient experienced knee arthritis six months before the first appointment and had cutaneous *vulgaris psoriasis* in elbow and knee only (Psoriasis Area Severity Index (PASI) = 1.8). The whole set of other pathologies leading to comorbidities and reduced BMI values were excluded. The patient was naïve of any conventional synthetic Disease Modifying Anti-Rheumatic Drug (CsDMARD), biological Disease Modifying Anti-Rheumatic Drug (bDMARD) or targeted synthetic Disease Modifying Anti-Rheumatic Drug (tsDMARD). Seven healthy volunteers with no sign of trabecular bone diseases or osteopenia (all males, mean age = 21.6 years [interquartile range (IQR) = 1 year], mean BMI = 21.32 kg/m$^2$ [IQR = 1.29 kg/m$^2$]) were included in the control group.

## MRI scanning

The patient and the volunteers underwent 7T MRI (MAGNETOM, Siemens Healthineers, Germany) of the knee joint (distal femur, proximal tibia and patella). All subjects were scanned using a 28-channel knee coil and a 3D gradient recalled echo sequence (3D GRE, TR/TE = 15/4.36 ms, flip angle = 12˚, bandwidth = 326 Hz/pixel, field of view = 180*180 mm, matrix = 768 x 768, in-plane voxel dimension 0.234 x 0.234 mm, slice thickness = 1.5 mm, 64 sagittal planes, acquisition time = 5 minutes 56 seconds). This protocol is similar to what has been previously used for knee scanning at 3T [28, 29]. The PsA patient was scanned once before treatment and once after one year of treatment. During MRI scanning, the patients' knee was immobilized by sandbags and secured by Velcro straps to avoid involuntary movements.

## PET scanning

As part of the usual follow-up procedure, the PsA patient underwent two CT/PET FNa scanning, once before treatment and once after one year of treatment. The sodium fluoride radiotracer (Cisnaf©) was administrated intravenously (3MBq/kg) and images were acquired 60 min after the injection on a Biograph 16 tomograph (Siemens, Healthineers, Germany), coupled to a low dose CT scanner with standard parameters (CT: 80 mA, 120 kV without contrast; 2 min per bed-PET- step of 15 cm) [30, 31]. CT/PET FNa images were iteratively reconstructed in a 128x128 matrix and 60 cm field of view, with and without attenuation correction in the transaxial, coronal and sagittal planes. The patient did not require special preparation. He was asked to be hydrated in order to activate the rapid washout of the radiotracer, to reduce the radiation dose and to improve the images quality.

## PET-MRI fusion

MR and CT/PET FNa images [30] were acquired using two different scanners. Given that bones were clearly visible in both CT and MR images, the four bones (femur, tibia, fibula, and patella) were used as landmarks for the registration of both images. More specifically, bones were delineated semi-automatically in each stack of images and linear affine registrations were computed independently between each bone using FSL-FLIRT [32]. Each local affine transformation was then merged into a global 3D deformation field through the implementation (described in [33]) of the log-euclidean poly-affine framework proposed by Arsigny et al. [34]. The resulting deformation field was used to overlay the PET maps on the highly resolved and contrasted 7T MR anatomical images as previously reported [35] (Fig 1).

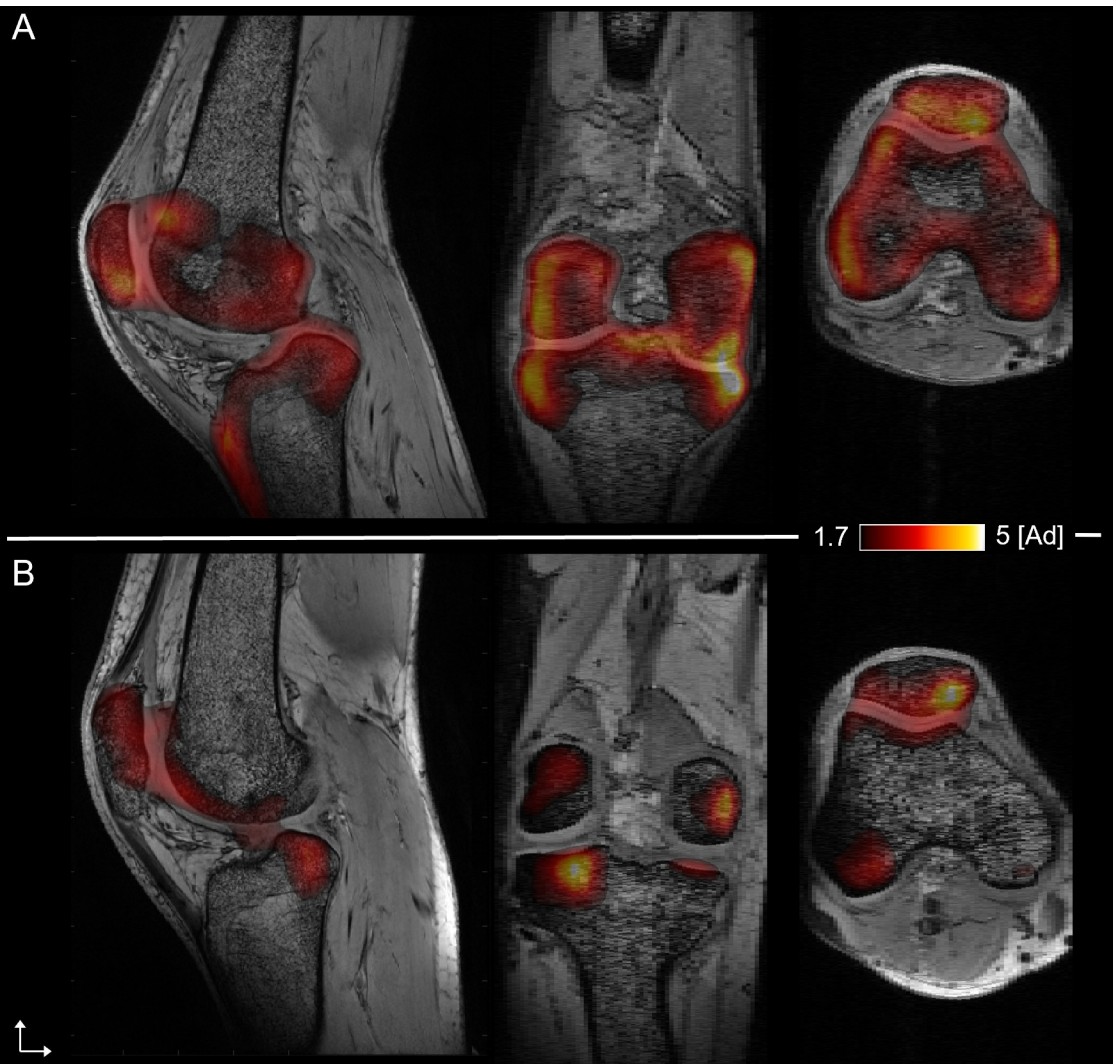

**Fig 1. Merged PET-UHF MRI.** Sagittal, coronal, and axial plane of merged PET-UHF MRI of the knee articulation of the patient before (A) and after (B) treatment by TNF-antibodies. "[Ad]" refers to a dimensional. Values higher or equal to 2.5 are considered indicative of "hypermetabolic" activity.

### PET-MR analysis

Fused PET-MR images were visually evaluated by an expert (SG) with the aim of identifying and localizing the hypermetabolic regions before and after the treatment. The visual inspection of fused images was crucial in order to identify the regions with hyperintense signals.

Bone volume fraction maps representing the relative volume of bone within each voxel were generated from the GRE images. The initial images were linearly scaled in order to cover the range from 0 (pure bone) to 255 (pure marrow) [36, 37]. In each image, distal femur, proximal tibia and patella were delineated using the Chan-Vese algorithm, which showed to be robust for the separation between bone, tendons and cartilage in the knee [38, 39]. The corresponding filled contours were used as masks on which a 10-pixels closing process was applied (2.34 mm) in all directions in order to eliminate all the cortical bone (Fig 2). Several region of interests (ROI) where identified in different locations of the trabecular bone in order to fully investigate the trabecular network.

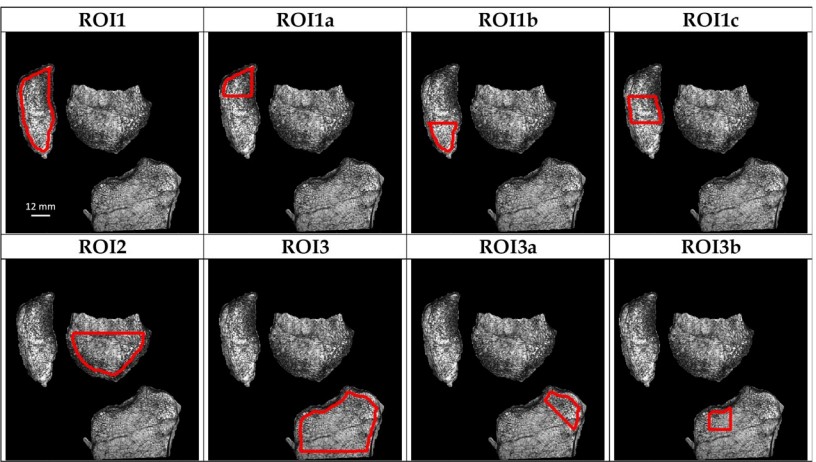

**Fig 2. ROIs identification.** PsA patient after treatment BVF maps showing the multiple ROIs identified in red.

**ROIs selection.** The ROIs selection was based on the PET-FNa results. Accordingly they were selected in regions with hyper-intense signals before the Adalimumab treatment and were selected in the same regions after the treatment regardless of the signal intensity.

*Patella.* The first set of ROIs (ROI1, ROI1a, ROI1b and ROI1c) were located in the patella region and referred respectively to the trabecular space of the whole patella, the upper and lower third of the trabecular region where the quadriceps and patellar tendons are respectively attached and the central third of the patella (Fig 2).

*Distal Femur.* ROI2 was located in the distal femur epiphysis as illustrated in Fig 2.

*Proximal Tibia.* The final set of ROIs (ROI3, ROI3a and ROI3b) were positioned in the proximal tibia. ROI3 refers to the trabecular space of the proximal tibia epiphysis. ROI3a represents the trabecular part of the tibia where the medial collateral ligand is attached and ROI3b represents the trabecular part of the tibia where there was no hypermetabolic activity on the basis of the PET FNa signal. (Fig 2).

**Bone microstructure evaluation.** To reduce the computational costs from the 3D ROIs, three 2D centrally located MRI planes were selected for each subject i.e. the image with the highest ROI surface together with the $N_{+1}$ and $N_{-1}$ images.

ROIs were then binarized using an automatic local thresholding as previously described [40] and three independent metrics were computed. The bone volume fraction (BVF) which refers to the ratio between bone and the total volume, the trabecular thickness (Tb.Th) and spacing (Tb.Sp). Tb.Th and Tb.Sp were extrapolated using iMorph [41] which can generate an aperture map (AM) derived from a distance transformation map. The AM was retrieved from the maximal balls diameter enclosed in the bone (Tb.Th) and in the marrow (Tb.Sp) phases (Fig 2). Finally the trabecular number (Tb.N) was computed as the ratio between the BVF and the Tb.Th.

Student's T-tests were used in order to assess the morphological parameters differences between the control group and the PsA patient before and after the TNF treatment. For each subject, three measurements were obtained for each metric and each ROI. A p-value lower than 0.01 was considered as significant.

**Standardized uptake values.** A semi-quantitative analysis of PET images was performed as previously described in order to generate the Standardized Uptake Values (SUV) [15, 30]. SUV were computed as the ratio between the signal intensity within each pixel of the image scaled to the concentration of the total injected radioactivity (3 MBq/Kg). The corresponding

results refer the pixel-based metabolic. A SUV of 2.5 or higher is generally considered to be indicative of an "hypermetabolic" region. Finally, mean and maximal values were computed within each ROI.

## Results

### PET-FNa

**Hypermetabolism evolution.** The visual inspection of the initial set of PET images showed intense polyarticular hyperintense signals preferentially involving the knees, the left hip, the right ankle, the elbows, and more moderately the spine, the feet and the hands. As illustrated in Fig 1, large hyperintensities were observed in the knee. The second set of PET image recorded after one year of treatment, showed an unequivocal reduction in most of the hypermetabolic regions affecting the joints of the axial and appendicular skeleton and more particularly the knee. The whole set of ROIs showed reduced hyperintensities whereas no more hyperintense signal was visible for ROI2 and ROI3b.

**SUV results.** SUV were quantified in all the knees ROIs before and after one year of treatment and the corresponding values are indicated in Table 1. Before the treatment, SUVmean was abnormal in 5 over 8 ROIs. The abnormal values were concentrated in all the patellar ROIs (2.7 ± 0.1) and ROI3a (2.8). SUVmax averaged over the whole set of ROIs was 3.67 ± 0.41. After the treatment, SUV were no longer larger than 2.5 in almost all the ROIs while the averaged SUVmax was also significantly reduced i.e. 2.86±0.86. Large SUV values (i.e. between 1.7 and 2.5) were still visible in all the patella ROIs and ROI3a (Table 1).

### MRI microarchitecture

Regarding the MRI-based micro-architecture measurements performed before the treatment, the patient was outside the control range for multiple metrics and multiple localizations (24 out of 32 measurements were statistically different from the controls). However, after one year of treatment the microarchitectural parameters differences between the PsA patient and the healthy references were reduced and the parameters were approaching or within the control range (only 9 out of 32 measurements were still statistically different than controls) (Table 2).

**Patella.** Before the treatment and considering the four ROIs delineated in the patellar region, BVF of the patient was always significantly lower as compared to controls with a mean difference of -23±10%. The Tb.Th difference was always below 5% (p>0.01 for all the four ROIs), with a general mean of 0.25±0.03 mm for the controls and 0.24±0.02 mm for the

**Table 1. SUV results before and after treatment for all identified ROIs.**

|  | Before Treatment | | After Treatment | |
|---|---|---|---|---|
|  | **SUVmean** | **SUVmax** | **SUVmean** | **SUVmax** |
| **ROI1** | 2.7±0.5 | 3.79 | 2.1±0.6 | 3.77 |
| **ROI1a** | 2.6±0.4 | 3.72 | 2.4±0.4 | 3.18 |
| **ROI1b** | 2.9±0.5 | 3.77 | 1.6±0.5 | 3.02 |
| **ROI1c** | 2.7±0.4 | 3.79 | 2.3±0.6 | 3.69 |
| **ROI2** | 1.9±0.5 | 3.34 | 1.2±0.6 | 3.32 |
| **ROI3** | 1.9±0.5 | 4.06 | 1.0±0.3 | 2.41 |
| **ROI3a** | 2.8±0.4 | 4.06 | 1.3±0.4 | 2.41 |
| **ROI3b** | 2.0±0.2 | 2.82 | 0.7±0.1 | 1.12 |

SUV mean (SUVmean) values are presented as mean ± SD and SUV maximum (SUVmax) values of the investigated ROIs before and after one year of treatment.

**Table 2. Microarchitecture characteristics per ROI.**

| | | | Controls | P. before treatment | P. after Treatment |
|---|---|---|---|---|---|
| Patella | ROI1 | BVF | 0.375±0.015 | 0.297±0.011 * | 0.373±0.016 |
| | | Tb.Th | 0.258±0.005 | 0.257±0.004 | 0.276±0.003 * |
| | | Tb.Sp | 0.429±0.065 | 0.643±0.036 * | 0.470±0.013 |
| | | Tb.N | 1.455±0.076 | 1.132±0.068 * | 1.347±0.008 |
| | ROI1a | BVF | 0.393±0.008 | 0.339±0.018 * | 0.401±0.010 |
| | | Tb.Th | 0.255±0.013 | 0.254±0.022 | 0.266±0.014 |
| | | Tb.Sp | 0.364±0.032 | 0.477±0.058 | 0.365±0.008 |
| | | Tb.N | 1.550±0.074 | 1.301±0.136 | 1.493±0.073 |
| | ROI1b | BVF | 0.355±0.035 | 0.222±0.064 * | 0.328±0.027 |
| | | Tb.Th | 0.261±0.010 | 0.250±0.015 | 0.285±0.004 * |
| | | Tb.Sp | 0.469±0.117 | 0.651±0.057 * | 0.532±0.064 |
| | | Tb.N | 1.366±0.114 | 0.994±0.090 * | 1.116±0.083 |
| | ROI1c | BVF | 0.377±0.015 | 0.295±0.026 * | 0.375±0.016 |
| | | Tb.Th | 0.207±0.008 | 0.213±0.005 | 0.225±0.003 * |
| | | Tb.Sp | 0.366±0.042 | 0.632±0.096 * | 0.424±0.005 |
| | | Tb.N | 1.746±0.250 | 1.409±0.119 * | 1.661±0.024 |
| Distal Femur | ROI2 | BVF | 0.354±0.048 | 0.257±0.015 * | 0.312±0.007 |
| | | Tb.Th | 0.261±0.005 | 0.260±0.006 | 0.269±0.006 |
| | | Tb.Sp | 0.516±0.140 | 0.769±0.025 * | 0.656±0.009 |
| | | Tb.N | 1.342±0.187 | 1.016±0.020 * | 1.173±0.054 |
| Proximal Tibia | ROI3 | BVF | 0.337±0.019 | 0.219±0.015 * | 0.256±0.012 * |
| | | Tb.Th | 0.266±0.011 | 0.245±0.004 * | 0.257±0.008 |
| | | Tb.Sp | 0.562±0.087 | 0.924±0.029 * | 0.866±0.053 * |
| | | Tb.N | 1.261±0.109 | 0.879±0.051 * | 0.985±0.043 * |
| | ROI3a | BVF | 0.381±0.009 | 0.307±0.016 * | 0.335±0.018 |
| | | Tb.Th | 0.258±0.008 | 0.260±0.009 | 0.267±0.012 |
| | | Tb.Sp | 0.426±0.060 | 0.594±0.012 * | 0.570±0.016 * |
| | | Tb.N | 1.468±0.073 | 1.185±0.047 * | 1.241±0.029 * |
| | ROI3b | BVF | 0.376±0.018 | 0.242±0.024 * | 0.285±0.013 * |
| | | Tb.Th | 0.220±0.015 | 0.192±0.005 * | 0.202±0.011 |
| | | Tb.Sp | 0.418±0.061 | 0.636±0.054 * | 0.539±0.033 |
| | | Tb.N | 1.689±0.148 | 1.255±0.183 * | 1.432±0.094 |

Data are presented as mean ± SD. "P." refers as patient. BVF: Bone volume fraction, Tb.Th: Trabecular Thickness, Tb.Sp: Trabecular Space, Tb.N: Trabecular number.
* indicates a statistically significant difference (p < 0.01) with the Healthy reference values.

patient. The Tb.Sp difference was statistically significant for ROI1, ROI1b and ROI1c but not for ROI1a with the patient having larger trabecular spaces as compared to controls and therefore a positive difference mean of 48±18%. Similar results were found for Tb.N and a significant difference was found for ROI1, ROI1b and ROI1c but not for ROI1a with a general mean difference of -21±5%.

Following the 12-month TNF treatment, most of the micro-architecture metrics but Tb.Th reversed to normal values. BVF increased in the four patella's ROIs thereby reducing the differences with controls to a non-significant mean value of -2±4%. Similar results were quantified for Tb.Sp and Tb.N with a non-significant difference with controls for any of the patella's ROIs and a new overall patient mean difference of 10±7% for Tb.Sp and -9±7% for Tb.N. On

**Fig 3. ROI1a extrapolated features box plot.** Box plot for each extrapolated feature for the control reference (Healthy), patient before (P_before) and after (P_after) one year of anti-TNFα treatment in the trabecular region where the quadricep tendon attaches the patella (ROI1a).

the contrary, after the treatment, Tb.Th became significantly larger with a significant difference (up to 9%) with controls and so for ROI1, ROI1b and ROI1c (Fig 3 and Table 2).

**Distal femur.**  In the distal femur (ROI2) the difference between the healthy reference and the patient before the treatment was more than 20% for all the parameters (-27% for BVF, 49% for Tb.Sp and -24% for Tb.N) except for Tb.Th for which the difference was less than 1%.

The image analysis after the treatment still showed increased BVF and Tb.N values while Tb.Sp values were reduced. The corresponding differences between the patient and the control values were -12%, -13% and +27% respectively. Similar to the results found in the patella, the Tb.Th increased becoming 3% thicker than controls. The difference between the control and the patient values after the treatment was statistically significant (p>0.01) for none of the micro-architectural parameters evaluated (Table 2).

**Proximal tibia.**  The three ROIs (ROI3, ROI3a and ROI3b) located in the proximal tibia region also showed statistically differences between patient and control values for the whole set of MRI metrics. The only normal value was found for Tb.Th in ROI3a. More particularly, the differences between the patient and the controls were -30±9% for BVF, 52±12% for Tb.Sp, -25±6% for Tb.N and -7±7% for Tb.Th.

After the 12 month-TNF treatment, the bone microstructure differences were reduced, although remaining statistically significant in most of the cases. For the BVF, the difference was reduced to -20±7% and remained statistically significant for ROI3 and ROI3b. The Tb.Th difference was also reduced to -3±6% thereby becoming not statistically significant for any of the three tibial ROIs. The Tb.Sp difference slightly decreased to 39±13% but remained statistically significant (p<0.01) for ROI3 and ROI3a but not for ROI3b. The Tb.N difference also decreased to -18±4% but remained statistically significant for ROI3 and ROI3a but not for ROI3b (Table 2).

## Discussion

In the present study, we assessed bone microarchitecture in a PsA patient in order to document the potential bone quality changes associated with his inflammatory status. We also assessed the microarchitecture modification resulting from a one-year anti-TNF treatment. We mainly found that PET-FNa/MRI showed a largely inflamed knee articulation with some specific hypermetabolic regions in the vicinity of ligament and tendons in the patella, the distal femur, and the proximal tibia. Microarchitectural changes quantified using UHF MRI were affecting the whole bone segments and were not localized within the hypermetabolic regions only. After a year of TNF treatment, the combined PET-UHF MRI approach showed highly reduced hypermetabolic regions and an improvement for most of the microarchitectural parameters and the BMI increased from 14.5 to 18.9 kg/m$^2$ reaching the normal range (18.5–24.9 kg/m$^2$) [42].

Before the treatment, all the microarchitecture metrics were significantly different with respect to the control values and so in at least one ROI. Using HR-pQCT on the distal radius of a group of 50 PsA patients and comparing the bone microarchitecture results to those from controls, Kocijan *et al.* reported significantly reduced BVF and Tb.N, increased Tb.Sp and almost constant Tb.Th [12]. Compared to our study, Kocijan *et al.* reported lower bone microstructure parameters differences between PsA patients and controls (-11.9%, -7.1%, +9.1%,-1.5% respectively for BVF, Tb.N, Tb.Sp and Tb.Th *vs.* an overall difference mean for all the ROIs analysed of -26% for BVF, -23% for Tb.N, +50% for Tb.Sp and -3% for Tb.Th). However, these discrepancies could be explained by the different anatomical investigated sites (distal radii *vs.* knee articulation) and by the age and body mass index of the PsA patients (51±13y, 27.9±5.1 kg/m$^2$ *vs.* 18y, 14.5 kg/m$^2$). Although previous DXA measurements have been controversial regarding BMD changes in PsA patients [8, 17, 18], our results further support those obtained using a radiating imaging technique and confirm abnormalities of trabecular bone in PsA patients so that osteoporotic changes might be expected in PsA.

In the field of rheumatologic inflammatory disorders, our study is the first to address the bone microarchitecture issue using UHF MRI, although previous studies involving the use of UHF MRI have reported promising results in osteoporosis [25–27, 43]. As an example, Chang *et al.* [25] found abnormal trabecular characteristics including BVF in the distal femur of subjects with fragility fractures whereas the DXA T-score was normal. Of interest, BVF, Tb.Sp and Tb.N were abnormal in the majority (7/8) of ROIs in the present study whereas Tb.Th was abnormal in a limited number (2/8) of ROIs. These results further support those previously reported by Kocijan *et al* [12] and Chang *et al.* [25] regarding the larger sensitivity of BVF, Tb.Sp and Tb.N to bone micro-architecture alterations as compared to Tb.Th. In fact, Kocijan *et al.* [12] reported no difference in Tb.Th between PA patients and healthy controls in distal radii while Chang *et al.* [25] found normal distal femur Tb.Th in patients with fragility fractures.

Trabecular abnormalities detected using UHF MRI were found in all the hypermetabolic regions detected using PET-FNa, showing that microarchitecture deterioration was affecting the whole bone segments. The PET analysis has been shown to reflect bone remodelling and has been used in several studies on osteoporosis [44–47]. In our case, PET-FNa allowed to localize specific ROIs characterized by elevated hypermetabolic activity before treatment and ROIs presenting partial or full remission after treatment.

After a year of anti-TNF treatment, the trabecular parameters clearly illustrated that the knee of the patient was in clinical remission from his PsA status. The trabecular parameters reversal might result from the decreased inflammatory status leading to a reduced osteoclastic bone resorption activity. In PsA, Hoff *et al.* [20] have showed that 24 weeks of Infliximab treatment can stop the bone loss. In multiple studies conducted in rheumatoid arthritis (RA) patients, the TNF blocking strategy has been associated with an increase of biological markers indicating bone formation and a decrease of those illustrating bone resorption [48–50]. In both RA and Ankylosing spondylitis (AS), the efficiency of anti-TNF agents on bone loss has also been confirmed through BMD measurements using DXA [49–53]. Our PET-FNa/MRI measurements also supported the efficiency of the anti-TNF strategy. In fact, UHF MRI allowed us to assess and quantify the microarchitectural parameters in the hypermetabolic ROIs assessed through the PET-FNa. In our study, UHF MRI showed an almost homogeneous microarchitecture deterioration before treatment and a partial or a complete remission after one year of treatment. These results are also in agreement with those previously reported as a result of bisphosphonates treatment in osteoporotic patients [45, 47].

A few limitations have to be acknowledged in the present study. Although, this preliminary study was conducted in a PsA patient, we have quantified morphological parameters in several

UHF MR images from 3 different bone segments (patella, distal femur, and proximal tibia) and using 8 different ROIs. Moreover, the results of the PsA patient were compared both temporally, i.e. before and after the treatment, and against the control group. One might wonder whether the reported changes are gender specific given that we assessed male subjects only and the inclusion of female subjects would be of interest. Additionally, it could be of interest to assess other bones regions with an elevated bone turnover such as the sacroiliac joint, spine and other peripheral joints. However, one has to keep in mind that the availability of dedicated coils for UHF MRI is rather reduced. One could also argue that partial volume effects might have biased the results. Such an effect can occur when pixels size in a given MR image is larger than the trabecular thickness (100 μm). Our protocol was similar to previously reported knee MRI acquisitions [28, 29]. The partial volume error if any was expected to be the same for all the MR images so that the comparison was still valid.

The investigation of bone microarchitecture in patients affected by PsA is of interest for a reliable assessment of bone quality, illness risk stratification and for the follow-up of therapeutic strategy. Up to now, PsA patients have been mainly treated using CsDMARD, bDMARD and tsDMARD [54] and the effects on bone microarchitecture have never been documented. However, the administration of anti-TNF may inhibit the osteoclastic action of bone resorption triggered by the inflammatory response. Moreover, the application of UHF MRI might be of high interest to investigate bone microarchitecture in the future for specific clinical situations.

## Author Contributions

**Conceptualization:** David Bendahan, Sandrine Guis.

**Data curation:** Enrico Soldati, Lucas Escoffier, Augustin C. Ogier.

**Formal analysis:** Enrico Soldati.

**Funding acquisition:** David Bendahan, Sandrine Guis.

**Investigation:** Enrico Soldati, Lucas Escoffier, Sophie Gabriel, Jean P. Mattei, Serge Cammilleri, David Bendahan, Sandrine Guis.

**Methodology:** Enrico Soldati, Sophie Gabriel, Jean P. Mattei, Serge Cammilleri, David Bendahan, Sandrine Guis.

**Project administration:** David Bendahan, Sandrine Guis.

**Resources:** Sophie Gabriel, Christophe Chagnaud, Jean P. Mattei, Serge Cammilleri, David Bendahan, Sandrine Guis.

**Software:** Enrico Soldati.

**Supervision:** David Bendahan, Sandrine Guis.

**Validation:** Enrico Soldati, Lucas Escoffier, David Bendahan, Sandrine Guis.

**Visualization:** Enrico Soldati, David Bendahan, Sandrine Guis.

**Writing – original draft:** Enrico Soldati, Lucas Escoffier.

**Writing – review & editing:** David Bendahan, Sandrine Guis.

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
