## [Decision Letter · Decision Letter 0]

15 Apr 2021

PONE-D-21-07600

Assessment of in vivo bone microarchitecture changes in an anti-TNFα treated psoriatic arthritic patient

PLOS ONE

Dear Dr. Enrico Soldati,

Thank you for submitting your manuscript to PLOS ONE. After careful consideration, we feel that it has merit but does not fully meet PLOS ONE’s publication criteria as it currently stands. Therefore, we invite you to submit a revised version of the manuscript that addresses the points raised during the review process.

We look forward to receiving your revised manuscript.

Kind regards,

Ewa Tomaszewska, DVM Ph.D

Academic Editor

PLOS ONE

Journal Requirements:

Reviewers' comments:

Reviewer's Responses to Questions

**Comments to the Author**

1. Is the manuscript technically sound, and do the data support the conclusions?

Reviewer #1: Yes

Reviewer #2: Yes

2. Has the statistical analysis been performed appropriately and rigorously? 

Reviewer #1: Yes

Reviewer #2: Yes

3. Have the authors made all data underlying the findings in their manuscript fully available?

Reviewer #1: Yes

Reviewer #2: Yes

4. Is the manuscript presented in an intelligible fashion and written in standard English?

Reviewer #1: Yes

Reviewer #2: Yes

5. Review Comments to the Author

Reviewer #1: This is a very interesting work. In my opinion it highlights a result that could have enormous application value in clinical practice. Therefore some "inaccuracies" need to be corrected:

1. In the paragraph Introduction, the Authors introduce the subject making some statements, some of which in my opinion, need to be reviewed.

First of all we know that sites highlighted by ultrasound may be clinically non-active. So this should be specified in the text : The most commonly involved sites by ultrasound assessment are………………(5).

2. In addition, it is not appropriate to define oligoarthritis, polyarthritis, dactylitis and enthesitis as symptoms as these are clinical presentations.

3. In order to give greater strength to their work, the Authors should remember that psoriatic arthritis has peculiar effects on the bone and that the prevalence of osteoporosis in this condition is currently subject of wide debate. In fact, in psoriatic arthritis, close to erosive manifestations we also observe productive manifestations ranging from periostitis to ankylosis. The greatest complication of osteoporosis in clinical practice are fragility fractures which represent a significant clinical aspect. In this regard I would recall the work of Del Puente A and co-workers published in the Journal of Rheumatology in 2015 ( J. Rheumatol Suppl.2015 Nov;93:36-39).

Finally,the patient studied must be accurately described.

1. What kind of psoriatic arthritis did he have: axial, peripheral or an overlap subset (axial and peripheral)?

2. Given the young age of the patient, it is necessary to specify how long psoriasis and arthritis had started.

3. What was the extent of psoriasis (calculated by PASI)?

Reviewer #2: The study title is focused on the bone microarchitecture changes in psoriatic arthritic patient but the data is focused on the male population. This may be additional limitation because there are differences in the architecture and metabolism in male and female bones. Images were acquired in the arthritic psoriasis regions. Usually there are other high bone turnover regions that can be used to monitor bone quality changes. My proposition is to consider future study in typical regions to confirm the thesis that changes are not only connected with the disease involved regions. Because of the UHF MR pixel size additional studies with microCT should be considered. There is no data of the psoriasis intensity on screening.

6. PLOS authors have the option to publish the peer review history of their article (what does this mean?). If published, this will include your full peer review and any attached files.

Reviewer #1: No

Reviewer #2: No

---

## [Author Response · Author response to Decision Letter 0]

28 Apr 2021

Detailed answers to Reviewer 1

This is a very interesting work. In my opinion it highlights a result that could have enormous application value in clinical practice. Therefore some "inaccuracies" need to be corrected:

In the paragraph Introduction, the Authors introduce the subject making some statements, some of which in my opinion, need to be reviewed.

• Point 1. First of all we know that sites highlighted by ultrasound may be clinically non-active. So this should be specified in the text : The most commonly involved sites by ultrasound assessment are………………(5).

We have now specified that ultrasound can detect the most common PsA sites as described by Tang et al. (1). These sites could be clinically active or non-active. However, the most commonly involved sites assessed by ultrasound are clinically active (lines 66-69). 

• Point 2. In addition, it is not appropriate to define oligoarthritis, polyarthritis, dactylitis and enthesitis as symptoms as these are clinical presentations.

According to the reviewer’s suggestion, the word “symptoms” has been replaced by “clinical presentations” (lines 69-74) (2,3).

• Point 3. In order to give greater strength to their work, the Authors should remember that psoriatic arthritis has peculiar effects on the bone and that the prevalence of osteoporosis in this condition is currently subject of wide debate. In fact, in psoriatic arthritis, close to erosive manifestations we also observe productive manifestations ranging from periostitis to ankylosis. The greatest complication of osteoporosis in clinical practice are fragility fractures which represent a significant clinical aspect. In this regard I would recall the work of Del Puente A and co-workers published in the Journal of Rheumatology in 2015 ( J. Rheumatol Suppl.2015 Nov;93:36-39).

This issue related to fragility fractures occurrence in PsA patients has been addressed in lines 74-78. We also discussed the issue of osteoporosis prevalence in PsA (4,5) (lines 74-78).

• Point 4. Finally, the patient studied must be accurately described.

What kind of psoriatic arthritis did he have: axial, peripheral or an overlap subset (axial and peripheral)?

Given the young age of the patient, it is necessary to specify how long psoriasis and arthritis had started.

What was the extent of psoriasis (calculated by PASI)?

Additional information about the PsA patient has been added. We mentioned that the patient was affected by axial and peripheral psoriatic arthritis with the first symptoms reported 6 months before the first medical appointment. Moreover, cutaneous vulgaris psoriasis was detected in elbow and knee with a Psoriasis Area Severity Index (PASI) = 1.8 (lines 108-112).

References

1. Tang Y, Cheng S, Yang Y, Xiang X, Wang L, Zhang L, et al. Ultrasound assessment in psoriatic arthritis (PsA) and psoriasis vulgaris (non-PsA): which sites are most commonly involved and what features are more important in PsA? Quant Imaging Med Surg. gennaio 2020;10(1):86–95. 

2. Coates LC, Helliwell PS. Psoriatic arthritis: state of the art review. Clin Med. febbraio 2017;17(1):65–70. 

3. Ritchlin CT, Colbert RA, Gladman DD. Psoriatic Arthritis. Longo DL, curatore. N Engl J Med. 9 marzo 2017;376(10):957–70. 

4. Del Puente A, Esposito A, Costa L, Benigno C, Del Puente A, Foglia F, et al. Fragility Fractures in Patients with Psoriatic Arthritis. The Journal of Rheumatology Supplement. 1 novembre 2015;93(0):36–9. 

5. Attia EAS, Khafagy A, Abdel-Raheem S, Fathi S, Saad AA. Assessment of osteoporosis in psoriasis with and without arthritis: correlation with disease severity: Assessment of osteoporosis in psoriasis. International Journal of Dermatology. gennaio 2011;50(1):30–5. 

 

Detailed answers to Reviewer 2

The study title is focused on the bone microarchitecture changes in psoriatic arthritic patient but the data is focused on the male population. This may be additional limitation because there are differences in the architecture and metabolism in male and female bones. Images were acquired in the arthritic psoriasis regions. Usually there are other high bone turnover regions that can be used to monitor bone quality changes. My proposition is to consider future study in typical regions to confirm the thesis that changes are not only connected with the disease involved regions. Because of the UHF MR pixel size additional studies with microCT should be considered. There is no data of the psoriasis intensity on screening.

According to the reviewer’s suggestion, we added as a limitation the fact that the study was conducted in male only and that due to architecture and metabolism differences between male and female, future studies in female and in other bone regions could be of interest (lines 361-365). We mentioned that cutaneous vulgaris psoriasis was present in elbow and knee with a Psoriasis Area Severity Index (PASI) = 1.8 (lines 108-112).

---

## [Decision Letter · Decision Letter 1]

4 May 2021

Assessment of in vivo bone microarchitecture changes in an anti-TNFα treated psoriatic arthritic patient

PONE-D-21-07600R1

Dear Dr. Enrico Soldati,

We’re pleased to inform you that your manuscript has been judged scientifically suitable for publication and will be formally accepted for publication once it meets all outstanding technical requirements.

Kind regards,

Ewa Tomaszewska, DVM Ph.D

Academic Editor

PLOS ONE

Additional Editor Comments (optional):

Reviewers' comments:

Reviewer's Responses to Questions

**Comments to the Author**

1. If the authors have adequately addressed your comments raised in a previous round of review and you feel that this manuscript is now acceptable for publication, you may indicate that here to bypass the “Comments to the Author” section, enter your conflict of interest statement in the “Confidential to Editor” section, and submit your "Accept" recommendation.

Reviewer #1: All comments have been addressed

Reviewer #2: All comments have been addressed

2. Is the manuscript technically sound, and do the data support the conclusions?

Reviewer #1: Yes

Reviewer #2: Yes

3. Has the statistical analysis been performed appropriately and rigorously? 

Reviewer #1: Yes

Reviewer #2: Yes

4. Have the authors made all data underlying the findings in their manuscript fully available?

Reviewer #1: Yes

Reviewer #2: Yes

5. Is the manuscript presented in an intelligible fashion and written in standard English?

Reviewer #1: Yes

Reviewer #2: Yes

6. Review Comments to the Author

Reviewer #1: the Authors have taken on board all the comments and amended the text according to them. The text has improved and is fully in response to expectations.

Reviewer #2: I recommend to accept this manuscript. Data presented corectly. Answers satisfy the reviewer. Additional informations were placed in the right form.

7. PLOS authors have the option to publish the peer review history of their article (what does this mean?). If published, this will include your full peer review and any attached files.

Reviewer #1: No

Reviewer #2: No

---

## [Editor Report · Acceptance letter]

10 May 2021

PONE-D-21-07600R1 

Assessment of *in vivo* bone microarchitecture changes in an anti-TNFα treated psoriatic arthritic patient 

Dear Dr. Soldati:

I'm pleased to inform you that your manuscript has been deemed suitable for publication in PLOS ONE. Congratulations! Your manuscript is now with our production department. 

Kind regards, 

on behalf of

Prof. Dr. Ewa Tomaszewska 

Academic Editor

PLOS ONE